# The Neonatal Environment and Health Outcomes (NEHO) Birth Cohort Study: Behavioral and Socioeconomic Characteristics and Drop-Out Rate from a Longitudinal Birth Cohort in Three Industrially Contaminated Sites in Southern Italy

**DOI:** 10.3390/ijerph18031252

**Published:** 2021-01-30

**Authors:** Silvia Ruggieri, Sabina Maltese, Gaspare Drago, Fabio Cibella, Simona Panunzi

**Affiliations:** 1Institute for Biomedical Research and Innovation, National Research Council of Italy, 90146 Palermo, Italy; silvia.ruggieri@irib.cnr.it (S.R.); sabinamaltese@libero.it (S.M.); gaspare.drago@irib.cnr.it (G.D.); 2Institute for System Analysis and Computer Science—BioMatLab, National Research Council of Italy, 00168 Rome, Italy; simona.panunzi@biomatematica.it

**Keywords:** industrially contaminated sites, birth cohort study, maternal exposure, fetal exposure, socioeconomic status, lifestyles

## Abstract

Pregnant women living in industrially contaminated sites (ICSs) are exposed to environmental contaminants through different pathways, and thus children’s health may be affected by pollutants. We created the Neonatal Environment and Health Outcomes (NEHO) longitudinal birth cohort in three ICSs in the Mediterranean area of southern Italy, collecting comprehensive information on personal data and lifestyles by questionnaire. Through multiple correspondence analysis, we identified possible clusters of enrolled women, and a neural network classifier analysis (NNCA) was performed to identify variables capable of predicting the attrition rate of the study. NEHO recruited 845 mother–child pairs over two years. The mothers’ mean age was 31.1 ± 5.2 SD years. We found significant differences in socioeconomic status (SES) among the three evaluated ICS, and an overall 11.1% prevalence of mothers who actively smoked during pregnancy. Active smoking during pregnancy was strongly associated with the lowest socioeconomic level (*p* < 0.0001). By means of the NNCA, we found that smoking during pregnancy and the lowest education level characterized the cluster with the highest attrition rate (*p* < 0.001). Our results demonstrate that reason for public health concern still exists regarding smoking during pregnancy and that SES influences both lifestyles, producing negative pregnancy outcomes and a higher survey attrition rate.

## 1. Introduction

Industrially contaminated sites are a serious public health concern, and there is growing focus on their impact on the environment and human health. The World Health Organization (WHO) defines contaminated sites as “*Areas hosting or having hosted human activities which have produced or might produce environmental contamination of soil, surface or groundwater, air, food chain, resulting or being able to result in human health impacts*” [1].

Living in proximity to industrially contaminated sites (ICSs), and thus being exposed to increased concentrations of environmental hazards, along with disadvantaged social and economic conditions, result in an increased incidence of diseases during childhood [2] and adulthood [3,4].

There are many ICSs in Europe [5], and in Italy a total of 57 ICSs were officially defined in 2009 as national priority contaminated sites (NPCSs) for environmental remediation.

A systematic literature review conducted by De Sario and colleagues showed that most of the available studies on highly polluted areas are descriptive (32.5%), while case–control and cohort studies represent less than 10% of the literature—of these, only 10 out of 762 peer-reviewed papers are based on birth cohorts [6].

The International Centre of Advanced Study in Environment, Ecosystem and Human Health (CISAS) project aimed at investigating environmental pollution and its connection with the ecosystem and human health, monitoring pollutant concentrations in human tissues and also performing in vitro/ex vivo experiments [7,8,9,10]. The CISAS project also included the extensive characterization of entire environmental matrices as well as food chains in three ICSs in southern Italy [11]. In Italy, in the context of the CISAS project, the Neonatal Environment and Health Outcomes (NEHO) cohort was built for recruiting mother–child pairs living in three different NPCSs, along with pregnant women living in surrounding areas with similar geographic and socio-demographic characteristics. The NEHO cohort is the first Italian cohort specifically created for the epidemiological evaluation of highly contaminated areas [12].

Fetuses and infants are highly vulnerable to toxicants. Epidemiological studies have shown that exposure to environmental stressors during in utero and perinatal developmental stages results in an increased incidence of childhood diseases [3,13,14,15]. Growth is most rapid in utero and during early childhood, and at these stages, environmental contaminants may produce the most damaging effects [3,16,17].

In the last decade, increasing public health concern has focused on the complex pathways of intrauterine growth and infancy development, during which exposure to toxicants should be avoided to give newborns the chance to start life as healthy as possible [18]. Many factors can play an important role in determining children’s health during early life, through a combination of genetic, environmental, and social factors, including maternal health during pregnancy [19].

Thus, the study of the effects of environmental exposures is an important environmental public health issue including (i) the increasing health risk of people living in ICSs [20], (ii) the influence of early-life exposures on the risk of both childhood and adult diseases [17] and metabolic profile alterations [21], and (iii) the ability of many pollutants to cross the placental membrane and interfere with fetal development [22].

Most of the known environment–health relationships in epidemiology are those in which high levels of exposure have been observed. However, less is known regarding the chronic low-dose exposure of vulnerable populations, such as children. In fact, chronic exposure to various chemicals may result in a number of adverse health outcomes after an extended time period.

Pregnant women living in highly contaminated sites can be exposed through different pathways, and children’s health may be affected by a wide range of chemicals [23,24]. Thus, the study of fetal growth and birth outcomes has become an important emerging field of environmental epidemiology, and birth cohorts provide an excellent opportunity to investigate the associations between early-life environmental exposures and children’s health in people living in ICSs [25].

The NEHO birth cohort aims at evaluating the role of environmental exposures (including physical, chemical, and psychosocial factors), lifestyles, and socioeconomic factors on fetal development, children’s early lives, and health outcomes in a population living near ICSs [12].

The quality of life and the health of a population living in an area strongly depends on the quality of the environment in a complex interaction among social, economic, and cultural factors [26,27,28]. Thus, in order to investigate the relationships among the exposure of pregnant women living in highly contaminated sites, their lifestyle, and socioeconomic status (SES), the NEHO birth cohort collected information on health status, smoking habits, diet, drug use, physical activity, occupational exposure, lifestyle, stress factors, and sociodemographic characteristics by means of web-based questionnaires.

The present work describes the two-year NEHO recruitment, the major characteristics of the enrolled cohort composed by 845 pregnant women, and the distribution of gestational and delivery characteristics. Finally, we aimed at describing the subset of mothers who dropped out of the study, also investigating the determinants of the attrition rate in the recruited women that was experienced in such a socioeconomic context.

## 2. Materials and Methods

### 2.1. Study Population

Recruitment started in January 2018 and ended in January 2020. The NEHO cohort enrolled, on a voluntary basis, 845 pregnant women living in the 3 NPCSs of Crotone, Augusta-Priolo, and Milazzo-Valle del Mela (hereafter referred to as Milazzo) in the Mediterranean area of southern Italy, along with pregnant women living in surrounding areas presenting similar geographic and sociodemographic characteristics. Pregnant women were recruited when admitted to the maternity units of the public hospitals of 4 cities: the “G. Fogliani” Hospital in Milazzo (for the Milazzo NPCS), the General Hospital of Lentini and the “Umberto I” Hospital in Syracuse (for the Augusta-Priolo NPCS), and the “San Giovanni di Dio” Hospital in Crotone. In each maternity unit, trained personnel provided a project leaflet to women attending the antenatal clinic. If the woman showed interest in the project, the personnel explained the study, also providing informed consent for the pregnant woman and her husband. In order to limit possible bias due to the self-selection of enrolled women and exclusive recruitment in public hospitals, we organized periodic meetings in birthing classes that also involved general practitioners. More details are available in a previous document [12].

On the basis of general inclusion criteria [12], we enrolled healthy pregnant women from the 34th week of pregnancy. Inclusion criteria are detailed in the Appendix A. After inclusion in the study, mothers were asked to fill out the baseline “first part” and “second part” questionnaires, collecting information on maternal health and lifestyle during the gestational and pre-gestational periods, respectively. After delivery, mothers were followed up at 6, 12, and 24 months by means of questionnaires collecting detailed information on the health and development status of their newborns; maternal work habits; use of electronic devices at home; dietary habits (validated food frequency questionnaires were used), including data on the type of water consumed and the place of fish, meat, and vegetable purchases; food packaging; and use of plastic dishes, detergents, and so on. The “baseline first part” questionnaire was administered during an interview carried out by qualified personnel. The “baseline second part” and all the follow-up questionnaires were filled in through the NEHO platform by means of a web interface. Information provided through the questionnaires is described in detail in Appendix A.

Moreover, at delivery, a form including general and clinical information about the type of delivery and birth outcomes (i.e., weight, length, head circumference) was filled in by healthcare personnel. Maternal blood samples were collected at enrollment and at delivery, when cord blood and placental samples were also collected [12]. Biological samples and their downstream application are described in Appendix A.

All questionnaire data were pseudonymized and stored on protected servers using ID tracking numbers, as were relevant bio-samples.

### 2.2. The Analyzed Questions

Aimed at describing the enrolled cohort as well as the subset of mothers who dropped-out of the study, this work presents the analysis of a subset of questions from the “baseline first part” questionnaire. This is divided into 8 sections and includes 122 global questions concerning personal data, sport activities, smoking habits, medical history, environmental and socioeconomic conditions, work activities, lifestyles, diet, alcohol consumption, and reproductive history.

We examined characteristics and socio-environmental factors relevant to both the mother and the father, along with some birth outcomes (i.e., gestational age, type of delivery, newborn weight, length, and head circumference).

Regarding the qualitative variables, some categories were merged: for “educational level”, the “elementary school” and “middle school” levels were unified into the “middle school or lower title” category. The “time to pregnancy” variable, originally coded into 4 categories (“less than 6 months”, “between 6 and 11 months”, “1 to 2 years”, and “more than 2 years”), was recoded into 2 categories by merging the first 2 into “less than 1 year” and the last 2 into “more than 1 year”. As concerns the variable “type of birth”, “spontaneous vaginal delivery or turning” and “operative vaginal” categories were considered as “vaginal”, while “scheduled caesarean” and “emergency caesarean” were recorded in the category “caesarean”.

Following the U.S. Institute of Medicine’s (IOM) recommendations [29], gestational weight gain (GWG) was defined as “*superior*”, “*recommended*”, or “*inferior*” on the basis of pre-pregnancy body mass index (BMI): “*recommended”* weight gain ranged from 12.5 to 18 kg for underweight women (UW), from 11.5 to 16 kg for normal weight women (NW), from 7 to 11.5 kg for overweight women (OW), and from 5 to 9 kg for obese women (OB).

### 2.3. Statistics

For continuous variables, one-way ANOVA was used to test differences among women living in the 3 NPCSs using Tukey’s test for post hoc comparisons. A chi-squared test or Fisher’s exact test, when appropriate, was used to study the association between “site” (NPCS) and categorical variables. Linear regression models were built for evaluating the relationships between continuous dependent variables (e.g., BMI) and quantitative or qualitative predictors (e.g., the 3 NPCSs, hospital units, age, or BMI). Logistic regression was applied to assess any type of relationship between binary dependent variables (e.g., type of birth: “vaginal” vs. “caesarean”) and qualitative or quantitative predictors. A multiple correspondence analysis (MCA) was used to identify possible patterns (groups) of enrolled women according to the categories of the studied variables by performing a dimensionality reduction and individuating those hidden dimensions (linear combination of the original variable categories) able to explain the data variability.

A neural network classifier analysis was performed to identify variables able to predict dropouts (binary output variable) by including all the information retrieved from the questions considered (qualitative and quantitative variables). The link between the set of input features (I) and the outcome (O) was chosen to be linear. The training set used to build the model was set to 70% of the entire sample; the rest of the sample was used as a testing set. Only one hidden layer (H) between input and output variables was considered. A cross-validation procedure was adopted to determine the optimal number of nodes in the hidden layer. Olden’s connection weights algorithm was used to evaluate the importance of input variables by considering weight signs and magnitudes.

A confusion matrix was used to describe the performance of the classification model. It is reported as a 2 × 2 table with the predicted values (dropout/no dropout) in the rows and the true classification in the columns.

Continuous variables are reported as mean ± SD, while categorical variables are reported as numbers and percentages. All *p*-values have an exploratory scope and must be considered as purely indicative of a possible difference among NPCSs or of a possible association among the considered factors. All tests were conducted at a nominal alpha error of 0.05, and no correction for multiple comparisons was performed. All the analyses were performed in R [30].

### 2.4. Ethics

The NEHO study was conducted in accordance with the Declaration of Helsinki, and the protocol was approved by the 3 ethics committees responsible for the involved NPCSs: the Ethics Committee of the University Hospitals of Messina for the NPCS of Milazzo (18 September 2017, no. 9/2017), the Ethics Committee “Catania 2” for the NPCS of Augusta-Priolo (11 July 2017, no. 38/2017/CECT2), and the Ethics Committee of Calabria Region for the NPCS of Crotone (20 July 2017, no. 173). Each participant read the information sheet and signed the informed consent form. Copies of participants’ information sheets are available at the study’s website (www.neho.it). Each participant was clearly informed that she could leave the project by sending an email to the NEHO website mail address.

## 3. Results

The NEHO cohort included 845 women, 561 (66.4%) from the Augusta-Priolo site, 167 (19.8%) from Crotone, and 117 (13.8%) from Milazzo. A total of 418 (49.5%) births occurred in the Syracuse unit and 143 (16.9%) in Lentini (for the Augusta-Priolo site), while 167 (19.8%) and 117 (13.8%) occurred in the Crotone and Milazzo hospitals, respectively. Table 1 reports the characteristics of the mothers included in the study. Mean age was 31.1 ± 5.2 and was not significantly different among the three sites (*p* = 0.13). Similarly, the distribution of marital status (never married, married, separated/divorced) was not significantly different (*p* = 0.13), with the greatest percentage of married women (63.4%) followed by never-married women (34.9%).

### 3.1. Socioeconomic and Behavioral Variables

Educational qualification was different in the three sites (*p* < 0.001), with the greatest percentage of graduated mothers in Milazzo at 36.7%, versus 22.2% and 25.4% in Augusta Priolo and Crotone, respectively. Conversely, in Milazzo, there was a smaller number of people with the lowest educational level (10.3%) than in the other two sites—28.3% in Augusta-Priolo and 24.2% in Crotone. These figures are in line with the working conditions during pregnancy—65.0% of women living in Milazzo worked until the 28th week of gestation versus 28.2% of women living in Augusta-Priolo and 38.8% of women resident in Crotone (*p* < 0.0001). In fact, the association between working during pregnancy and educational level was significant—the logistic model produced an odds ratio (OR) of working during pregnancy for mothers with the highest educational level being about 12 times (*p* < 0.0001) that of mothers with the lowest educational level.

While “active smoking” was not significantly different among the three sites (*p* = 0.64), the relationship between “active smoking” during pregnancy and educational level was significant, with an OR of 0.14 (*p* < 0.001) of smoking during pregnancy for women with the highest educational level (degree or higher qualification) and an OR of 0.34 (*p* < 0.001) for women with the middle educational level (high school) with respect to women with the lowest educational level. “Passive smoking” was, instead, significantly different among the three sites (*p* < 0.0001), with a small percentage of women living in Augusta-Priolo subjected to passive smoke—8.6% versus 21.0% and 29.1% in Crotone and Milazzo, respectively.

Alcohol consumption was also significantly different among the three sites (*p* < 0.0001)—1.2% in Augusta-Priolo, versus 10.0% and 13.7% in Crotone and Milazzo, respectively.

### 3.2. General Health Status and Pregnancy

Baseline (before pregnancy) BMI was not significantly different among the three sites (*p* = 0.32), but it decreased significantly with increasing educational level—the linear regression model provided a β = −1.1 ± 0.4 kg/m^2^ (*p* = 0.003) for secondary educational level and a β = −1.96 ± 0.4 kg/m^2^ (*p* < 0.0001) for women graduates versus the lowest class of education. Weight gain was significantly greater in the Milazzo sample than in the Crotone sample—12.9 ± 3.9 kg versus 11.3 ± 4.4 (*p* = 0.006 from Tukey’s post hoc analysis). Figure 1 shows the distribution of weight gain in the categories “*superior*”, “*recommended*”, and “*inferior*”, with respect to BMI baseline values, within each site. Moreover, an inverse relationship was found between weight gain and baseline BMI (β = −0.19 ± 0.03 kg, linear regression model, *p* < 0.0001)—the Cochran–Mantel–Haenszel test for the association between classes of BMI (“underweight”, “normal-weight”, “overweight”, “obesity”) and classes of recommended weight gain (“*superior*”, “*recommended*”, “*inferior*”), when corrected for the site strata factor, turned out to be significant (*p* < 0.0001). In all sites, a larger-than-recommended weight gain was observed in women with larger BMI (*p* < 0.05 from post hoc analysis).

The percentage of intended pregnancies was 78%, and it was significantly different among the three sites (*p* < 0.0001)—61.2% in Crotone, 82.9% in Milazzo, and 82.7% in Augusta-Priolo. For women wishing to become pregnant, the time to become pregnant was not significantly different among the sites (*p* = 0.16)—90.6% of pregnancies occurred within one year and 9.4% after more than one year.

Table 2 reports the characteristics of the 845 fathers of the NEHO cohort. The average age was 34.6 ± 6.3, with a slight, although significant, older age in men from Milazzo (*p* = 0.02). As for the mothers, the fathers living in Milazzo presented a higher educational level—21.4% graduates versus 12.9% and 15.2% in Augusta-Priolo and Milazzo, respectively. Paternal BMI was not significantly different among the three sites, but the relationship between BMI and educational level was significant also among fathers (linear regression model)—β = −1.16 ± 0.31 (*p* = 0.0002) and β = −1.32 ± 0.43 (*p* = 0.002) for the high school level and the degree level, respectively, compared to the lowest educational level (secondary school or lower qualification).

The mothers living in Augusta-Priolo underwent their first ultrasound scan at 8.1 ± 2.6 weeks of gestation, while mothers living in Crotone underwent the exam earlier, at 7.2 ± 2.1 weeks, and mothers in Milazzo at 7.6 ± 2.1 weeks (*p* < 0.001, post hoc comparisons are reported in Figure 2). The mean gestational week at birth was 39.5 ± 1.2 and was significantly different among the three sites (*p* < 0.0001), with a smaller value in the Augusta-Priolo sample at 39.4 ± 1.2, versus 39.6 ± 1.1 and 39.8 ± 1.0 in Crotone and Milazzo, respectively (*p* = 0.024 and *p* < 0.001, respectively, from Tukey’s post hoc comparisons; Figure 2).

Participation in birthing classes was significantly different among the three sites (*p* = 0.001)—the highest percentage of women who attended courses was recorded in Augusta-Priolo with 44.3%, in comparison with 30.3% and 32.5% of women from Crotone and Milazzo, respectively. The distribution of the variable “type of birth” was significantly different among the three sites (*p* < 0.0001), with 43.7% of caesarean sections in Augusta-Priolo, 33.3% in Milazzo, and 17.4% in Crotone. BMI and age were tested as possible predictors of “type of birth” in a multivariable logistic model—OR of age and BMI for cesarean birth were 1.04 (*p* = 0.002) and 1.06 (*p* = 0.0009), respectively. As concerns the variable NPCS, the odds ratio was 3.86 (*p* < 0.0001) for women living in Augusta-Priolo and 2.50 (*p* = 0.001) for women living in Milazzo, with respect to women living in Crotone. In a univariable model where “hospital unit” was considered as a possible predictor of the “type of birth”, the odds ratio related to the two units of the Augusta-Priolo site were greater than one and significant with respect to the Crotone unit—the risk of caesarean sections was more than four times greater (*p* < 0.0001) for the Syracuse hospital and about double for the Lentini hospital (*p* = 0.018).

The sample of newborns was composed of 845 newborns (54.4% males and 45.6% females), and these values were not different among the three sites. In our cohort, we did not record any still birth or neonatal death. While birth weight did not differ, newborns in Augusta-Priolo presented with a smaller length (*p* < 0.0001), and newborns in Crotone presented with a smaller head circumference (*p* < 0.0001, Table 3). Birth weight was, however, significantly associated with the “*recommended or inferior*” weight gain, and this relationship remained statistically significant after controlling for NPCS. Birth weight increased by 79.3 ± 34.8 g for newborns of mothers who gained more weight with respect to the “*recommended or inferior*” one (*p* = 0.023, linear regression model). Similarly, the relationship between newborn length and mother’s height was significant after controlling for NPCS and father’s height variables; birth length increased by 0.05 ± 0.009 cm with an increase of one unit (i.e., 1 cm) of the mother’s height (*p* < 0.0001, linear regression model).

### 3.3. Attrition Rate

The overall attrition rate of the study was 15.4% (130 of 845 recruited mother/child pairs)—33 (25.4% of the total dropouts) from the Crotone site, 25 (19.2%) from Milazzo, and 72 (55.4%) from Augusta-Priolo (45 (34.6%) from the Syracuse unit, 27 (20.8%) from the Lentini unit). Figure 3 reports the distribution of women who dropped out for each NPCS; the association between “dropout” and NPCS was significant (*p* = 0.014)—odds ratios of leaving the study for mothers living in Crotone and Milazzo were 1.67 (*p* = 0.026) and 1.85 (*p* = 0.018) with respect to mothers living in Augusta-Priolo.

Table 4 reports the characteristics of the sub-sample of women who dropped out—mean age was 30 ± 6.2 years, not significantly different among the three sites (*p* = 0.70), but the relationship between the “dropout” variable and age was significant, with an OR of 0.95 (*p* = 0.009, logistic regression) for each additional age year. As for the total sample, in the dropout sample, the educational qualification was statistically different among the three sites (*p* = 0.009), with a greater percentage of graduate mothers in Milazzo at 16.0% versus 13.9% and 9.0% in Augusta-Priolo and in Crotone, respectively. Conversely, in Milazzo, there was a smaller number of people with the lowest educational level (20.0%) with respect to the other two sites, where the percentages were 58.3% (Augusta-Priolo) and 54.6% (Crotone). Moreover, in this case, educational qualification was in line with the working condition in pregnancy—52.0% of women living in Milazzo worked until the 31st week of gestation versus 16.7% of women living in Augusta-Priolo and 12.1% of women resident in Crotone (*p* < 0.001). The relationship between “dropout” and educational level was significant, with an OR of 0.20 (*p* < 0.001) of leaving the study for women with the highest educational level (degree or higher qualification) and 0.29 (*p* < 0.001) for women with the middle educational level (high school) with respect to women with the lowest educational level. Table 5 reports the characteristics of the fathers related to the “dropout” sub-sample—the average age was 33.4 ± 6.4. The sub-sample of newborns, as reported in Table 6, was composed of 55.4% males and 44.6% females; gender distribution was not different among the three sites. As in the case of the total sample, the birth weight did not differ, while newborns in Augusta-Priolo presented with a shorter length (*p* < 0.0001), and newborns in Crotone presented with a smaller head circumference (*p* < 0.001).

In the overall sample, 138 women were excluded from the MCA due to the presence of missing values in at least one of the considered variables: marital status, working condition, first ultrasound, preparation course, NPCS, age, BMI, weight gain, education level, active smoking, passive smoking, alcohol consumption, intended pregnancy. The analysis therefore included a total of 707 women. Four dimensions were sufficient to describe the sample variability (49.1% of total inertia) (Figure 4, panel a).

The MCA highlighted the presence of three clusters (Figure 4, panel b):Cluster 1, composed of 181 mothers (25.6%) and characterized by high percentages of women residing in Milazzo, married, with the highest educational qualification; by high frequencies of working women with alcohol consumption habits and not exposed to both active and passive smoke, as well as by a high percentage of women who attended pre-birth courses (blue dots).Cluster 2, composed of 401 mothers (56.7%) and characterized by a high percentage of women living in Augusta-Priolo, non-working with a middle educational level, looking for a pregnancy and not attending pre-birth courses. The cluster is also composed of high percentages of women not exposed to active and passive smoke and not consuming alcohol (green dots).Cluster 3, composed of 125 mothers (17.7%) and characterized by a high percentage of women living in Crotone, never married, with the lowest educational qualification; by high frequencies of non-working women who did not seek a pregnancy and who did not attend pre-birth courses. The cluster also presented a high percentage of women exposed to active and passive smoke, with alcohol consumption habits (red dots).

A total of 108 mothers out of 707 included in the MCA dropped out the study—14 (13.0%) belonged to the first cluster, 54 (50.0%) to the second cluster, and 40 (37.0%) to the third. The relationship between “dropout” and cluster variable was significant—the odds ratios for dropping out the study were 1.86 (*p* = 0.049) and 5.61 (*p* < 0.001) for women classified in the second and third clusters, respectively, compared to the first cluster.

To predict dropouts (output variable), the neural network classifier analysis included the following input variables: NPCS, age, education level, marital status (never married, married, separated/divorced), working condition, active smoking, and intended pregnancy. The cross-validation procedure identified three nodes as the optimum node number for the hidden layer. The top panel in Figure 5 (panel a) shows the neural network plot with 10 input nodes, a three-node hidden layer, and the output node. Two bias nodes, embodying the meaning of the intercept in a linear model, were necessary for data interpretation. The bottom panel in Figure 5 (panel b) shows the importance of the variable in the estimated model.

Living in Crotone, exposure to active smoking, the lowest educational qualification, and working and marital status were all factors positively associated with dropping out. Conversely, age, trying to get pregnant, never having been married, and having a high level of qualification presented a negative association with dropping out. The top panel of Figure 6 (panel a) shows the confusion matrix including the number of true and false positive/negative dropouts generated by the network.

The accuracy rate obtained for the testing sample was 87.1%. The bottom panel in Figure 6 (panel b) shows the confusion matrix for each NPCS—in Augusta-Priolo and in Milazzo, about 90% of women were classified as true “no-dropout”, while about 10% of mothers were classified as false no-dropout.

## 4. Discussion

NEHO is the first Italian longitudinal birth cohort to evaluate mother–child pair characteristics in a population living in ICSs, including detailed information from questionnaires and collecting biological samples from participating mother–child pairs. In this work, we describe the distribution of maternal and child characteristics of the NEHO birth cohort, whose recruitment started on January 2018 and ended on January 2020, also evaluating the possible determinants of withdrawal from the study in such a population. Data were collected through interviews and self-administered questionnaires.

Pregnancy and birth cohort studies are aimed at studying the impacts of exposures during early life and development of the child, usually collecting biological material from mothers and children to evaluate biomarkers of exposure to contaminants, early effects, or susceptibility [31].

In the last few decades, numerous birth cohorts have been created worldwide to evaluate environmental exposures [32,33]. Conversely, in Italy, where 57 NPCSs exist, in spite of increasing public health research, there are few, if any, such studies. Environmental exposures in NPCSs, along with the increasing risk of early-life diseases, make it necessary to evaluate the impact of these environmental factors on this high-risk population.

The NEHO birth cohort recruited 845 mothers aged 31.1 ± 5.2 years and their children. Here, we describe the characteristics of the mothers included in the study. The average value of age and marital status was not significantly different in the three evaluated sites. Conversely, educational level was different among the three sites. In fact, the portion of study participants with a high school level of education was higher in Milazzo than in Augusta-Priolo and Crotone. This result is also reflected in working condition—most of the study participants living in Milazzo were employed and worked until the 28th week of gestation. Moreover, the association between working condition during pregnancy and educational level was significant as women with a higher educational level were more likely to work until the end of their pregnancy.

The relationship between “active smoking” during pregnancy and educational level was significant, with a lower risk of smoking during pregnancy for women with the highest educational level (OR 0.14) with respect to women with a secondary or lower level (reference category). In fact, active maternal smoking during pregnancy—reported by 11.1% of pregnant women in the overall sample—was more frequent among mothers with a lower educational level. Moreover, passive smoke was significantly different among the three sites, showing a higher percentage of women subjected to passive smoke in Milazzo. Similarly, we found that alcohol consumption was significantly different in the three areas, with the highest percentage of women consuming alcohol in Milazzo. Alcohol consumption and smoking during pregnancy are associated with many adverse birth outcomes, such as spontaneous abortions, preterm births, small for gestational age, low birth weight, ectopic pregnancies, placenta previa, and fetal alcohol spectrum disorders [34,35,36,37,38]. Despite several studies recommending that pregnant women and parents should be more informed about the negative impact of tobacco smoke exposure and alcohol consumption during pregnancy and on birth outcomes [39,40,41], this demonstrates that a public health concern still exists regarding this matter.

We evaluated the effect of the mothers’ educational levels on BMI before pregnancy. Baseline BMI was not different among the three sites (*p* = 0.32), but it decreased significantly with increasing educational level. In 2009, the IOM issued an update of guidelines for weight gain in pregnancy referring to the WHO’s BMI categories and identifying targets for GWG for OB, OW, NW, and UW BMI categories [29]. Our data show that the 2009 IOM recommendations for weight gain were not followed by many enrolled pregnant women. The inability to control weight gain during pregnancy is particularly evident in OW and OB women. Our results show that women with higher pre-pregnancy BMI during the gestational period gained more weight than recommended, with a significant association between classes of baseline BMI (UW, NW, OW, OB) and classes of recommended weight gain (“*superior*”, “*recommended*”, “*inferior*”). In all sites, weight gain was significantly higher in the overweight and obese women sample.

Information on gestational age—according to the last menstrual period—was obtained from data collected by midwives after delivery, and this was significantly different among the three sites, with a smaller average value in the Augusta-Priolo sample (*p* < 0.0001). Moreover, this study evaluated the type of delivery—the analyses indicate that delivering in the Syracuse and Lentini hospitals involved a risk of caesarean section more than four times and two times greater than being born in the Crotone hospital, respectively. This suggests that differences exist among medical staffs in decision making relevant to pregnant women.

Birth cohort studies are designed to study the effects of early exposures during the development of the child, and maintaining a stable cohort requires a constant effort. In these studies, women are usually approached during pregnancy and then mother–child volunteer pairs are followed up for years; thus, clear and accurate information on the project and study protocols is crucial to make the enrollment possible and the follow-up participation rate stable, which is a very important factor in attaining reliable study results [42]. We tried to investigate the characteristics of participants dropping out of the study—no statistically significant differences were found among the three sites in terms of age, educational level, and working condition of the subjects who withdrew from the study. The highest percentage of subjects who dropped out of the study was recorded among mothers living in Crotone and Milazzo, compared to those living in Augusta-Priolo. Our analysis shows that mothers who dropped out of the project were younger and had an educational level lower than mothers who continued with the follow-up. Accordingly, among the factors able to influence participant withdrawal, it was shown that low–middle educational level and low SES influenced lower response and higher drop-out rates [43,44].

In order to characterize the factors that could produce an increased risk for dropping out, we performed an MCA analysis to identify groups of mothers with similar characteristics on the basis of the answers provided to the questionnaire. The MCA analysis identified three different clusters in which enrolled women with similar profiles were distributed. The first cluster was constituted by a high percentage of women residing in the Milazzo NPCS with the highest educational qualification, high frequency of working, with alcohol consumption habits, and not having been exposed to either active or passive smoke. The second and the third clusters described the profile of most of the mothers participating in the project and were characterized by women with a low–middle educational level.

The distribution of the dropouts in the three clusters identified from the MCA analysis revealed a larger concentration of women in the second cluster and, mostly, in the third cluster. The socio-demographic characteristics of mothers in the NEHO cohort reflect a population with low–medium socio-economic conditions for the majority of the participants. Almost all study subjects were characterized by low–middle educational level and were unemployed, in accordance with the very low rates of employment in southern Italy [45]. The proportion of study participants with a high educational level who dropped out is very low, suggesting that such enrolled women are more willing to participate in studies than their less educated counterparts.

As for the reasons related to dropping out, from the neural network analysis, we saw that smoking during pregnancy and the lowest level of education—indicators of a lower socioeconomic status—were the most important variables associated with a higher probability of withdrawal from the study. The results related to smoking during pregnancy could be interpreted as a lower sensitivity and attention of the mothers to their own pregnancy status. Low educational level is in line with a lower awareness of the relationship between possible environmental problems and health conditions, which could lead to the attribution of little importance to scientific approaches. The relationship between dropping out and the two sites of Milazzo and Crotone that emerged from this analysis is in the same direction as that observed in Figure 3. Conversely, as expected, a higher educational level and an intended pregnancy decreased the propensity to drop out of the study. Older age decreased the probability of leaving the study, highlighting a greater involvement and perhaps concern of older mothers about the future of their newborns.

### Strengths and Limitations

The strengths of the NEHO study include the socio-demographic characteristics of the participants living in the ICSs; the relatively large sample size; the possibility of evaluating environmental exposures through questionnaires and biological matrices such as maternal and cord blood and placenta, collected at delivery; and hair samples, urine, and children’s toenails collected at multiple time points across the follow-up period starting at age two. We collected detailed information about pregnancy conditions and delivery health outcomes, and followed up at 6 months, 12 months, and 24 months. The questionnaires were designed concerning a possible comparison with other cohort studies and information provided in the context of the CISAS Project. Moreover, although in Italy there are many large sample size birth cohorts [46,47], there are no studies that investigate populations living within or close to NPCSs.

On the other hand, the weaknesses of this paper are that more than half of the participants were enrolled in the hospitals of the Augusta-Priolo NPCS and that mothers who dropped out of the study can no longer participate. Moreover, the recruitment—obviously performed on an exclusively voluntary basis—could have been biased by the same sociocultural level of participating women, producing different interests toward the health-related aspects of daily living in highly contaminated areas.

## 5. Conclusions

In conclusion, in this work, we present the NEHO birth cohort characteristics while also attempting to identify the factors influencing the attrition rate of our study. NEHO cohort enrolled mother–child pairs residing in disadvantaged areas with a low socioeconomic status, a poorly studied condition in the context of Italian and European birth cohorts. We believe that the cohort design, through the collection of biological material, the periodic collection of questionnaires, and performing clinical evaluations during the follow-up, will allow us to describe the complexity of chronic multi-toxic exposures, the socio-economic determinants, the behavior of the maternal lifestyle during pregnancy, and their combined effects on the health of the populations residing in the NPCSs. We also hope that a new network of collaboration between the European birth cohorts can be built, especially for industrially polluted areas.

## Figures and Tables

**Figure 1 ijerph-18-01252-f001:**
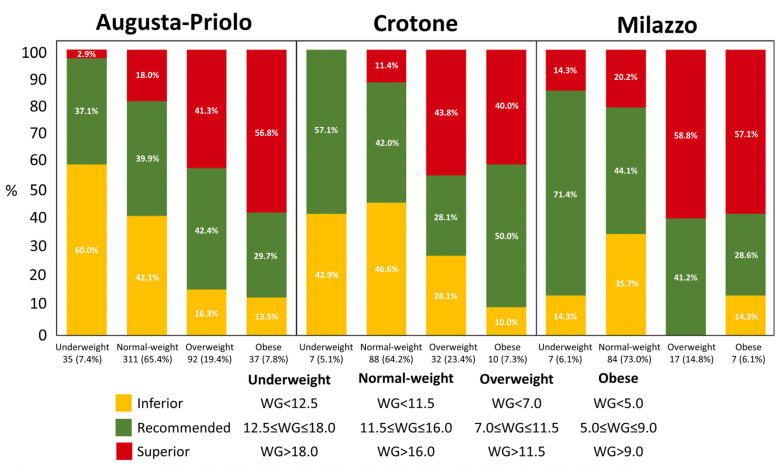
Distribution of weight gain in the categories “superior”, “recommended”, and “inferior” with respect to body mass index (BMI) baseline values within each national priority contaminated site.

**Figure 2 ijerph-18-01252-f002:**
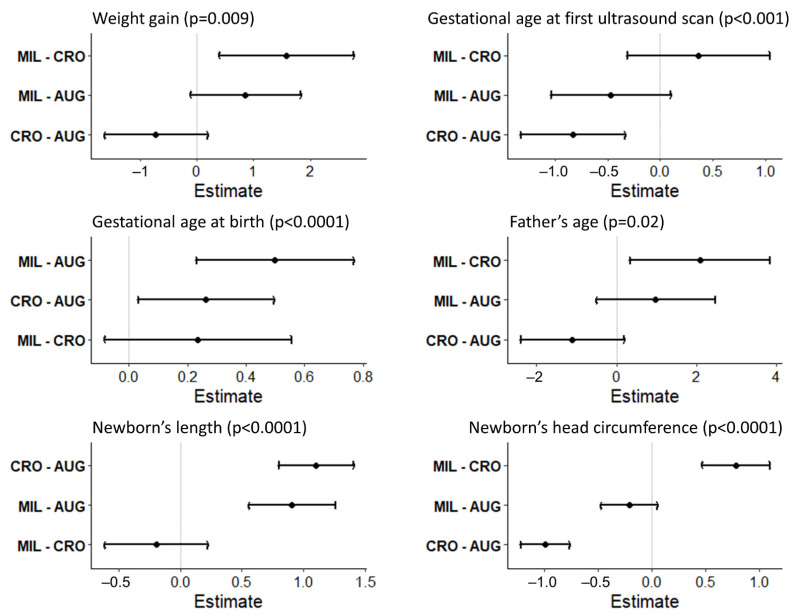
Tukey’s post hoc multiple comparisons among the three national priority contaminated sites (NPCSs). Six variables relevant to gestational period and birth outcomes are evaluated. MIL: Milazzo NPCS; AUG: Augusta-Priolo NPCS; CRO: Crotone NPCS.

**Figure 3 ijerph-18-01252-f003:**
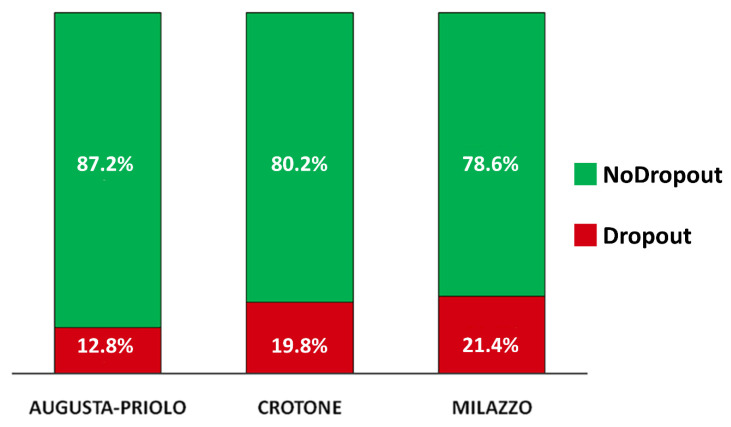
Distribution of dropouts within each national priority contaminated site.

**Figure 4 ijerph-18-01252-f004:**
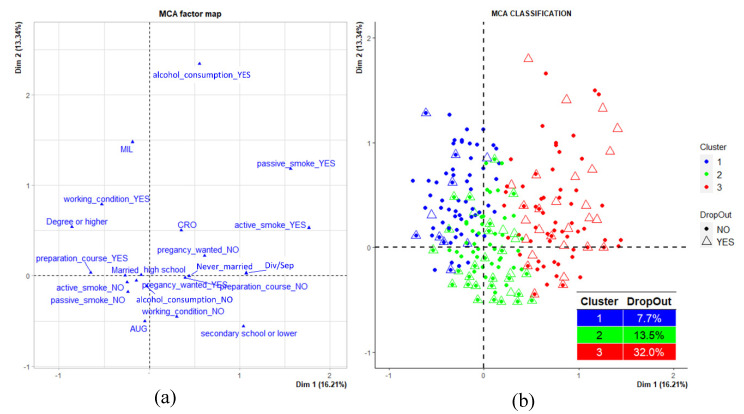
Multiple correspondence analysis (MCA): four dimensions described the sample, and three clusters were identified. (**a**) Dimensions describing the sample variability; (**b**) Distribution of drop out in the three clusters in MCA classification.

**Figure 5 ijerph-18-01252-f005:**
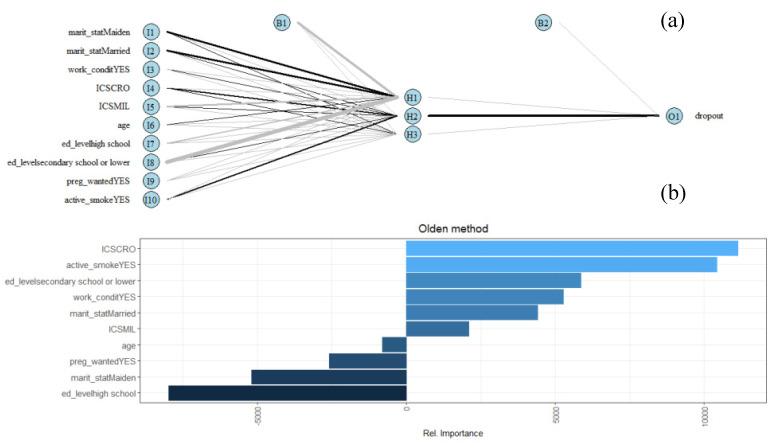
Neural network classifier analysis including the following input variables: Industrially Contaminated Site (ICS), age, educational level, marital status (never married, married, separated/divorced), working condition, active smoking, and pregnancy wanted. (**a**) The neural network plot with 10 input nodes, a three-node hidden layer, and the output node; (**b**) importance of the variables in the estimated model.

**Figure 6 ijerph-18-01252-f006:**
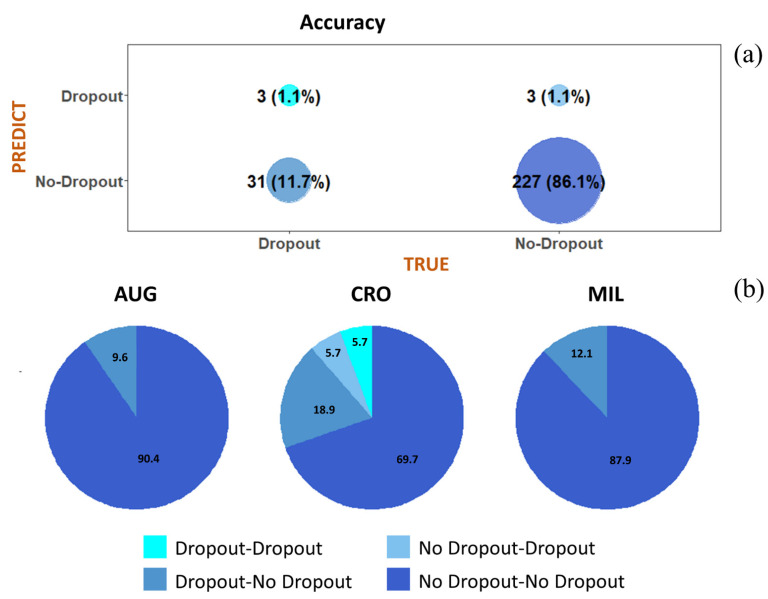
Confusion matrix to describe the performance of the neural network classifier model for total testing sample and within each evaluated site. (**a**) Confusion matrix including the number of true and false positive/negative dropouts generated by the network; (**b**) confusion matrix for each National Priority Contaminated Site.

**Table 1 ijerph-18-01252-t001:** Characteristics of the enrolled women in the total sample and in each national priority contaminated site (NPCS): continuous variables are reported as mean ± SD, while categorical variables are reported as numbers and percentages. Significant *p*-values are in bold. For each categorical variable, in parentheses, the total number of responders is indicated.

Characteristics	Augusta-Priolo		Crotone		Milazzo		*p*-Value ^†^	Total Sample (*n* = 845)	
	Mean	SD	Mean	SD	Mean	SD		Mean	SD
**Age (years)**	30.9	5.2	31.0	5.6	32.0	4.6	0.13 *	31.1	5.2
**Marital Status**	**No. (559)**	**%**	**No. (167)**	**%**	**No. (117)**	**%**	0.13 **	**No. (843)**	**%**
Never married	208	37.2	48	28.7	38	32.5		294	34.9
Married	341	61.0	117	70.1	76	65.0		534	63.4
Divorced	3	0.5	2	1.2	2	1.7		7	0.8
Separated	7	1.3	0	0.00	1	0.8		8	0.9
**Educational Level**	**No. (559)**	**%**	**No. (165)**	**%**	**No. (117)**	**%**	**<0.001** **	**No. (841)**	**%**
Secondary school or lower qualification	158	28.3	40	24.2	12	10.3		210	25.0
High school	277	49.5	83	50.4	62	53.0		422	50.2
Degree or higher qualification	124	22.2	42	25.4	43	36.7		209	24.8
**Working Condition**	**No. (556)**	**%**	**No. (165)**	**%**	**No. (117)**	**%**	**<0.0001** **	**No. (838)**	**%**
YES	157	28.2	64	38.8	76	65.0		297	35.4
NO	399	71.8	101	61.2	41	35.0		541	64.6
**Average week of gestation in which they stopped working**	**Mean**	**SD**	**Mean**	**SD**	**Mean**	**SD**		**Mean**	**SD**
	27.8	9.7	24.2	11.7	27.6	9.9	0.05 *	27	10.3
**Reasons** ^#^	**No. (148)**	**%**	**No. (61)**	**%**	**No. (76)**	**%**		**No. (285)**	**%**
Reached the legal deadline for maternity	75	50.7	22	36.1	8	10.5		105	36.8
Disease/tiredness	11	7.4	9	14.7	6	7.9		26	9.1
Personal choice	53	35.8	27	44.3	65	85.5		145	50.9
Dismissal	3	2.0	1	1.6	0	0.00		4	1.4
Other	5	3.4	1	1.6	1	1.3		7	2.5
**Active Smoking**	**No. (559)**	**%**	**No. (166)**	**%**	**No. (117)**	**%**	0.64 **	**No. (842)**	**%**
YES	63	11.3	20	12.1	10	8.6		93	11.1
NO	496	88.7	146	87.9	107	91.4		749	88.9
	**Mean**	**SD**	**Mean**	**SD**	**Mean**	**SD**		**Mean**	**SD**
Average number of cigarettes per day in pregnancy	5.6	4.8	3.3	2.9	4.3	2.9	0.10*	5.0	4.4
Average number of cigarettes per day in first trimester	5.8	4.9	3.4	2.8	4.5	3.2	0.11*	5.1	4.4
Average number of cigarettes per day in second trimester	5.3	4.6	2.6	2.9	3.6	2.4	0.06 *	4.6	4.3
Average number of cigarettes per day in third trimester	5.1	4.3	2.8	2.9	3.6	2.3	0.12*	4.5	3.9
**Passive Smoking**	**No. (561)**	**%**	**No. (167)**	**%**	**No. (117)**	**%**	**<0.0001** **	**No. (845)**	**%**
YES	48	8.6	35	20.9	34	29.1		117	13.9
NO	513	91.4	132	79.1	83	70.9		728	86.1
**Consumption of Alcohol**	**No. (560)**	**%**	**No. (160)**	**%**	**No. (117)**	%	**<0.0001** **	**No. (837)**	**%**
YES	7	1.2	16	10.0	16	13.7		39	4.7
NO	553	98.8	144	90.0	101	86.3		798	95.3
	**Mean**	**SD**	**Mean**	**SD**	**Mean**	**SD**		**MEAN**	**SD**
**BMI (kg/m** ^**2**^ **)**	23.6	4.8	23.4	4.1	22.9	3.7	0.32 *	23.4	4.6
**Weight gain (kg)**	12.1	4.0	11.3	4.4	12.9	3.9	**0.009** *	12.1	4.1
**Intended Pregnancy**	**No. (561)**	**%**	**No. (165)**	**%**	**No. (117)**	**%**	**<0.0001** **	**No. (843)**	**%**
YES	464	82.7	101	61.2	97	82.9		662	78.5
NO	97	17.3	64	38.8	20	17.1		181	21.5
**Time to Pregnancy**	**No. (464)**	**%**	**No. (98)**	**%**	**No. (97)**	**%**	0.16 **	**No. (659)**	**%**
Less than one year	414	89.2	93	94.9	90	92.8		597	90.6
More than one year	50	10.8	5	5.1	7	7.2		62	9.4
**First Ultrasound**	**Mean**	**SD**	**Mean**	**SD**	**Mean**	**SD**	**<0.001** *	**MEAN**	**SD**
Week of gestation	8.1	2.6	7.2	2.1	7.6	2.1		7.8	2.4
**Preparation Course**	**No. (560)**	**%**	**No. (165)**	**%**	**No. (117)**	**%**	**0.001** **	**No. (842)**	**%**
YES	248	44.3	50	30.3	38	32.5		336	39.9
NO	312	55.7	115	69.7	79	67.5		506	60.1
**Birth Type**	**No. (561)**	**%**	**No. (167)**	**%**	**No. (117)**	**%**	**<0.0001** **	**No. (845)**	**%**
Vaginal	316	56.3	138	82.6	78	66.7		532	63.0
Caesarean section	245	43.7	29	17.4	39	33.3		313	37.0
	**Mean**	**SD**	**Mean**	**SD**	**Mean**	**SD**	**<0.0001** *	**MEAN**	**SD**
Week of gestation	39.4	1.2	39.6	1.1	39.9	1.0		39.5	1.1

^†^*p*-value computed for the differences between the sites; * one-way ANOVA; ** χ^2^ test; ^#^ multiple-choice question. Significant *p*-values are in bold.

**Table 2 ijerph-18-01252-t002:** Characteristics of fathers in the total sample and in each NPCS: continuous variables are reported as mean ± SD, while categorical variables are reported as numbers and percentages. Significant *p*-values are in bold. For each categorical variable, in parentheses, the total number of responders is indicated.

Characteristics	Augusta-Priolo		Crotone		Milazzo		*p*-Value	Total Sample (*n* = 845)	
	Mean	SD	Mean	SD	Mean	SD		Mean	SD
**Age (years)**	34.6	6.3	33.5	6.6	35.6	5.6	**0.02** *	34.6	6.3
**Marital Status**	**No. (453)**	**%**	**No. (166)**	**%**	**No. (116)**	**%**	0.28 **	**No. (735)**	**%**
Never married	166	36.6	47	28.3	38	32.8		251	34.1
Married	281	62.1	117	70.5	75	64.7		473	64.4
Divorced	2	0.4	1	0.6	2	1.7		5	0.7
Separated	4	0.9	1	0.6	1	0.8		6	0.8
**Educational Level**	**No. (556)**	**%**	**No. (164)**	**%**	**No. (117)**	**%**	**0.008** **	**No. (837)**	**%**
Secondary school or lower qualification	191	34.4	52	31.7	22	18.8		265	31.7
High school	293	52.7	87	53.1	70	59.8		450	53.8
Degree or higher qualification	72	12.9	25	15.2	25	21.4		122	14.5
**BMI (kg/m^2^)**	**Mean**	**SD**	**Mean**	**SD**	**Mean**	**SD**		**Mean**	**SD**
	26.0	3.9	25.9	3.7	25.3	2.6	0.15 *	25.9	3.7

*p*-value computed for the differences between the sites; * one-way ANOVA; ** χ^2^ test. Significant *p*-values are in bold.

**Table 3 ijerph-18-01252-t003:** Characteristics of the newborns in the total sample and in each NPCS: continuous variables are reported as mean ± SD, while categorical variables are reported as numbers and percentages. Significant *p*-values are in bold. For each categorical variable, in parentheses, the total number of responders is indicated.

Characteristics	Augusta-Priolo		Crotone		Milazzo		*p*-Value	Total Sample (*n* = 845)	
**Newborn Gender**	**No. (561)**	**%**	**No. (167)**	**%**	**No. (117)**	**%**	0.42 **	**No. (845)**	**%**
Male	302	53.8	98	58.7	60	51.3		460	54.4
Female	259	46.2	69	41.3	57	48.7		385	45.6
	**Mean**	**SD**	**Mean**	**SD**	**Mean**	**SD**		**Mean**	**SD**
**Weight**	3288.7	407.2	3335.6	389.6	3347.9	364.6	0.19 *	3306	398.5
**Length**	49.8	1.4	50.9	1.8	50.7	1.6	**<0.0001** *	50.1	1.6
**Head Circumference**	34.8	1.0	33.8	1.2	34.6	1.6	**<0.0001** *	34.6	1.2

*p*-value computed for the differences between the sites; * one-way ANOVA; ** χ^2^ test. Significant *p*-values are in bold.

**Table 4 ijerph-18-01252-t004:** Characteristics of the dropout mothers in the total subsample and in each NPCS: continuous variables are reported as mean ± SD, while categorical variables are reported as numbers and percentages. Significant *p*-values are in bold. For each categorical variable, in parentheses, the total number of responders is indicated.

Characteristics	Augusta-Priolo		Crotone		Milazzo		*p*-Value	Sub-Sample (*n* = 130)	
	Mean	SD	Mean	SD	Mean	SD		Mean	SD
**Age (years)**	29.6	6.1	30.3	7.3	30.7	4.6	0.70 *	30	6.2
**Marital Status**	**No. (72)**	**%**	**No. (33)**	**%**	**No. (25)**	**%**	0.06 **	**No. (130)**	**%**
Never married	29	40.3	7	21.2	12	48.0		48	37.0
Married	41	56.9	26	78.8	11	44.0		78	60.0
Divorced	1	1.4	0	0.00	1	4.0		2	1.5
Separated	1	1.4	0	0.00	1	4.0		2	1.5
**Educational Level**	**No. (72)**	**%**	**No. (33)**	**%**	**No. (25)**	**%**	**0.009** **	**No. (130)**	**%**
Secondary school or lower qualification	42	58.3	18	54.6	5	20.0		65	50.0
High school	20	27.8	12	36.4	16	64.0		48	36.9
Degree or higher qualification	10	13.9	3	9.0	4	16.0		17	13.1
**Working Condition**	**No. (72)**	**%**	**No. (33)**	**%**	**No. (25)**	**%**	**<0.001** **	**No. (130)**	**%**
YES	12	16.7	4	12.1	13	52.0		29	22.3
NO	60	83.3	29	87.9	12	48.0		101	77.7
**Average gestation week in which they stopped**	15.3 ± 9.3		32.0 ± 1.6		31.4 ± 8.2		**<0.001** *	25.1 ± 11.3	
**Reasons**	**No. (11)**	**%**	**No. (4)**	**%**	**No. (13)**	**%**		**No. (28)**	**%**
Reached the legal deadline for maternity	1	9.1	2	50.0	0	0.00		3	10.7
Disease/tiredness	2	18.2	1	25.0	0	0.00		3	10.7
Personal choice	4	36.4	1	25.0	13	100.0		18	64.3
Dismissal	0	0.00	0	0.00	0	0.00		0	0.00
Other	4	36.4	0	0.00	0	0.00		4	14.3
**Active Smoking**	**No. (72)**	**%**	**No. (33)**	**%**	**No. (25)**	**%**	0.51 **	**No. (130)**	**%**
YES	13	18.1	9	27.3	4	16.0		26	20.0
NO	59	81.9	24	72.7	21	84.0		104	80.0
	**MEAN**	**SD**	**Mean**	**SD**	**Mean**	**SD**		**Mean**	**SD**
Average number of cigarettes per day in pregnancy	6.3	4.0	3.4	3.0	4.6	2.5	0.18 *	5.1	3.6
Average number of cigarettes per day in first trimester	6.6	4.3	3.5	3.1	4.00	2.7	0.17 *	5.7	3.9
Average number of cigarettes per day in second trimester	6.0	4.2	3.1	3.4	4.8	2.6	0.26 *	4.8	3.8
Average number of cigarettes per day in third trimester	6.0	3.8	3.5	3.3	4.4	2.3	0.30	4.8	3.5
**Passive Smoking**	**No. (72)**	**%**	**No. (33)**	**%**	**No. (25)**	**%**	**0.0001** **	**No. (130)**	**%**
YES	9	12.5	14	42.4	12	48.0		35	26.9
NO	63	87.5	19	57.6	13	52.0		95	73.1
**Consumption of Alcohol**	**No. (72)**	**%**	**No. (32)**	**%**	**No. (25)**	**%**	**0.02** **	**No. (129)**	**%**
YES	2	2.8	5	15.6	4	16.0		11	8.5
NO	70	97.2	27	84.4	21	84.0		118	91.5
	**MEAN**	**SD**	**MEAN**	**SD**	**MEAN**	**SD**		**MEAN**	**SD**
**BMI (kg/m^2^)**	23.2	4.6	23.9	4.8	23.9	3.2	0.67 *	23.5	4.4
**Weight Gain (kg)**	12.3	4.1	12.1	4.6	12.2	3.4	0.99 *	12.2	4.1
**Intended Pregnancy**	**No. (72)**	**%**	**No. (33)**	**%**	**No. (25)**	**%**	0.67 **	**No. (130)**	**%**
YES	51	70.8	23	69.7	20	80.0		94	72.3
NO	21	29.2	10	30.3	5	20.0		36	27.7
**Time to Pregnancy**	**No. (51)**	**%**	**No. (23)**	**%**	**No. (20)**	**%**	0.82 **	**No. (94)**	**%**
Less than one year	46	90.2	21	91.3	17	85.0		84	89.4
More than one year	5	9.8	2	8.7	3	15.0		10	10.6
**First Ultrasound**	**MEAN**	**SD**	**MEAN**	**SD**	**MEAN**	**SD**	0.51 *	**MEAN**	**SD**
Gestation week	8.1	2.7	7.6	2.5	8.2	2.5		8.0	2.6
**Preparation Course**	**No. (72)**	**%**	**No. (33)**	**%**	**No. (25)**	**%**	0.70 **	**No. (130)**	**%**
Yes	23	31.9	8	24.2	6	24.0		37	28.5
No	49	68.1	25	75.8	19	76.0		93	71.5
**Birth Type**	**No. (72)**	**%**	**No. (33)**	**%**	**No. (25)**	**%**	0.29 **	**No. (130)**	**%**
Vaginal	41	56.9	23	69.7	18	72.0		82	63.1
Caesarean section	31	43.1	10	30.3	7	28.0		48	36.9
	**MEAN**	**SD**	**MEAN**	**SD**	**MEAN**	**SD**	0.42 *	**MEAN**	**SD**
Week of gestation	39.3	1.2	39.4	1.2	39.7	0.9		39.4	1.1

*p*-value computed for the differences between the sites; * one-way ANOVA; ** χ^2^ test. Significant *p*-values are in bold.

**Table 5 ijerph-18-01252-t005:** Characteristics of the dropout fathers in the total subsample and in each NPCS: continuous variables are reported as mean ± SD, while categorical variables are reported as numbers and percentages. Significant *p*-values are in bold. For each categorical variable, in parentheses, the total number of responders is indicated.

Characteristics	Augusta-Priolo		Crotone		Milazzo		*p*-Value	Sub-Sample (*n* = 130)	
	Mean	SD	Mean	SD	Mean	SD		Mean	SD
**Age (years)**	33.8	7.0	32.7	5.7	33.2	5.6	0.72 *	33.4	6.4
**Marital Status**	**No. (56)**	**%**	**No. (23)**	**%**	**No. (55)**	**%**	**0.03** **	**No. (114)**	**%**
Never married	21	37.5	7	21.2	13	52.0		41	35.9
Married	33	58.9	26	78.8	11	44.0		70	61.4
Divorced	0	0.00	0	0.00	1	4.00		1	0.9
Separated	2	3.6	0	0.00	0	0.00		2	1.8
**Educational Level**	**No. (69)**	**%**	**No. (33)**	**%**	**No. (25)**	**%**	0.51 **	**No. (127)**	**%**
Secondary school or lower qualification	34	49.3	17	51.5	8	32.0		59	46.5
High school	32	46.4	14	42.4	15	60.0		61	48.0
Degree or higher qualification	3	4.3	2	6.1	2	8.00		7	5.5
**BMI (kg/m^2^)**	**Mean**	**SD**	**Mean**	**SD**	**Mean**	**SD**		**Mean**	**SD**
	26.3	3.3	26.2	3.7	25.9	3.0	0.87 *	26.2	3.3

*p*-value computed for the differences between the sites; * one-way ANOVA; ** χ^2^ test. Significant *p*-values are in bold.

**Table 6 ijerph-18-01252-t006:** Characteristics of the dropout newborns in the total subsample and in each NPCS: continuous variables are reported as mean ± SD, while categorical variables are reported as numbers and percentages. Significant *p*-values are in bold. For each categorical variable, in parentheses, the total number of responders is indicated.

Characteristics	Augusta-Priolo		Crotone		Milazzo		*p*-Value	Sub-Sample (*n* = 130)	
**Newborn Gender**	**No. (72)**	**%**	**No. (33)**	**%**	**No. (25)**	**%**	0.57 **	**No. (130)**	**%**
Male	38	52.8	21	63.6	13	52.0		72	55.4
Female	34	47.2	12	36.4	12	48.0		58	44.6
	**Mean**	**SD**	**Mean**	**SD**	**Mean**	**SD**		**Mean**	**SD**
**Weight**	3286.5	409.2	3311.8	397.6	3410.7	313.9	0.39 *	3317	389.8
**Length**	49.6	1.5	51.0	1.8	50.8	1.6	**<0.0001** *	50.2	1.7
**Head Circumference**	34.7	1.0	33.7	1.2	34.5	1.5	**<0.001** *	34.4	1.2

*p*-value computed for the differences between the sites; * one-way ANOVA; ** χ^2^ test. Significant *p*-values are in bold.

## Data Availability

No applicable.

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
