# Peer review of "The Neonatal Environment and Health Outcomes (NEHO) Birth Cohort Study: Behavioral and Socioeconomic Characteristics and Drop-Out Rate from a Longitudinal Birth Cohort in Three Industrially Contaminated Sites in Southern Italy"

_ijerph, 2021, doi:10.3390/ijerph18031252_

Round 1

Reviewer 1 Report

General comments:

Very interesting and well written paper describing the enrollment and characteristics of the Neonatal Environment and Health Outcomes (NEHO) birth cohort. Such detail of the cohort will be invaluable for looking at effects of NPCS in future years. The characteristics of drop outs and predictions of factors for drop outs is often a concern in epi studies, which makes publication of this paper favourable.

It is noticable that the study did not include other more standard birth outcomes, that are often measured when looking at pollution effects on maternal health. E.g. birth weight- TLBW, PTLBW, still birth, neonatal deaths,  disability at birth, etc? This could limit the findings and application of such ecological cohort study. Should this be listed as a limitation? Will these be collected at future time points?

The findings of Fig 4b) are very interesting and show that those mothers with riskier behaviours had higher drop out. Could more be made of this finding in the discussion and abstract? Researchers looking for evidence of differential behaviours of drop outs would find this of great interest.

What are the implications for future studies and the effects of NPCS? What are the research questions, in relation to NPCS, that the birth cohort study hopes to address in the future?

Specific comments:

Line 104- Supp materials – refer to Table S2. Why were basic biomonitoring samples taken at 24 months only? Increasing frequency can show changes in exposure, and effects.

Line 113- it would be useful here to mention the method of recruitment, were mothers approached by HCW, or researchers, at what point of their interaction with health services (briefly only, and refer to Ruggieri, 2019, for more details).

Line 144: replace ‘As concerns’ with ‘Regarding....’

Line 154: add ’ ...gain’ to recommended weight ranged...?

Section 2.4 – how could participants drop out/ withdraw from the study?  

Line 242 – how many fathers in the sample?

Line 276, how many newborns were there, still births, neonatal deaths etc?

Line 480. Table ST2 suggests that child biological samples (urine, hair, nails) were not collected at multiple points, but at 0 and 24 months, contradictory to what is written here. What is the plan for future frequencies?

Author Response

Reviewer #1

General comments:

Very interesting and well written paper describing the enrollment and characteristics of the Neonatal Environment and Health Outcomes (NEHO) birth cohort. Such detail of the cohort will be invaluable for looking at effects of NPCS in future years. The characteristics of drop outs and predictions of factors for drop outs is often a concern in epi studies, which makes publication of this paper favourable.

R: We thank the Reviewer for her/his appreciation of our work

It is noticeable that the study did not include other more standard birth outcomes, that are often measured when looking at pollution effects on maternal health. E.g. birth weight- TLBW, PTLBW, still birth, neonatal deaths, disability at birth, etc? This could limit the findings and application of such ecological cohort study. Should this be listed as a limitation? Will these be collected at future time points?

R: In our cohort, 845 children are included. Among them, only 26 (3.1%) birthed before than 37th week of gestation. The 819 remaining (96.9%) all were born at term. Among those who were born at term, 12 (1.5%) had a birth weight lower than 2500g, while the remaining 807 (98.5%) were born with a birth weight higher than 2500g. We had only 26 preterm infants. At birth, we recorded all the available clinical variables: type of delivery, gender, gestational age, newborn’s weight, length and head circumference, Apgar scores.

The findings of Fig 4b) are very interesting and show that those mothers with riskier behaviours had higher drop out. Could more be made of this finding in the discussion and abstract? Researchers looking for evidence of differential behaviours of drop outs would find this of great interest.

R: Never-married women, lowest educational qualification, and smoking during pregnancy were factors producing a higher risk for drop out.

The last three paragraphs in Discussion section before section “4.1. Strengths and Limitations” were modified to strength this finding that is now better reported also in the Abstract.

What are the implications for future studies and the effects of NPCS? What are the research questions, in relation to NPCS, that the birth cohort study hopes to address in the future?

R: NEHO enrolled mother-child pairs residing in disadvantaged areas with a low socioeconomic status, a poorly studied condition in the context of Italian and European birth cohorts. We believe that the cohort design, through the collection of biological material, the periodic collection of questionnaires and performing clinical evaluations during the follow-up, will allow us to describe the complexity of chronic multi-toxic exposures, the socio-economic determinants, the behavior of the maternal lifestyle during pregnancy and their combined effects on the health of the populations residing in the NPCSs. We also hope that a new network of collaboration between the European birth cohorts can be built especially for industrially polluted areas.

Specific comments:

Line 104 – Supp materials – refer to Table S2. Why were basic biomonitoring samples taken at 24 months only? Increasing frequency can show changes in exposure, and effects.

R: The basic biomonitoring samples – aimed at establishing the possible exposure to pollutants of both the mother and the child during pregnancy – were obtained at enrollment (last weeks of the pregnancy) and at delivery, when maternal and cord blood were collected along with placental samples. We intend to follow children’s exposure during the life starting collecting biological samples at two years of life

Line 113: it would be useful here to mention the method of recruitment, were mothers approached by HCW, or researchers, at what point of their interaction with health services (briefly only, and refer to Ruggieri, 2019, for more details).

R: The method of recruitment was integrated in the first paragraph (subsection 2.1) of Materials and Methods section

Line 144: replace ‘As concerns’ with ‘Regarding....’

R: The expression was changed as requested

Line 154: add ’...gain’ to recommended weight ranged...?

R: The Reviewer is right: the missing word was added

Section 2.4 – how could participants drop out/withdraw from the study?

R: Participation was on a voluntary basis, so each participant was clearly informed that she can leave the project by sending an email to NEHO website mail address. The sentence was added at the end of subsection “2.4 Ethics”

Line 242 – how many fathers in the sample?

R: As per protocol, no mother/child pair was enrolled if newborn’s father did not sign the informed consent. That is, in formal terms, we had the same number (845) of mothers and fathers. However, information relevant to fathers is sometime missing (only mothers filled the questionnaire). Following Reviewer’s observation, we now indicate the number of available responses per each categorical variable in all the Tables.

Line 276, how many newborns were there, still births, neonatal deaths etc?

R: The information is now integrated in the last paragraph of results section

Line 480. Table ST2 suggests that child biological samples (urine, hair, nails) were not collected at multiple points, but at 0 and 24 months, contradictory to what is written here. What is the plan for future frequencies?

R: The Reviewers is right, the information provided in the first paragraph of section “4.1. Strengths and Limitations” was incorrect. Now it was corrected. Plan for future frequencies includes sampling of hair, urine and children’s toenails every two years starting at age two.

Reviewer 2 Report

Comments and suggestions for authors:

The authors created the Neonatal Environment and Health Outcomes (NEHO) longitudinal birth cohort in three ICSs in the Mediterranean area of southern Italy, collecting comprehensive information on personal data and lifestyles by questionnaire, to explore possible risk factors for negative pregnancy outcomes. The results suggested that SES influences both lifestyles known to produce negative pregnancy outcomes and a higher survey attrition rate. Our comments on this article are as follows:

1.The title of this article is not clear and specific, and readers don't know what the authors want to say? Abbreviations should not be used in the title, for even if an abbreviation is well known in one specialty, it could be confusing to readers from other specialties.

  1. The research goal of this paper is not clear.

3.If the author wants to study the impact of environmental pollution on adverse pregnancy outcomes, there are serious defects in the design of this article. For example, since the study population comes from three high pollution areas, the environmental pollutants in the air, water, soil, and food should be the main risk factors for adverse pregnancy outcomes. This study did not involve the impact of environmental pollutants on adverse pregnancy outcomes. Obviously, the conclusion is not reliable.

  1. Even if we don't talk about the impact of environmental pollutants on adverse pregnancy outcomes, the conclusions of this study are not innovative and belong to popular science knowledge.
  2. In the resultsection, each independent content should havea subtitle.
  3. The NEHO cohort included 845 women, 561 (66.4%) from the Augusta-Priolo site, 167 (19.8%) from Crotone, and 117 (13.8%) from Milazzo. Due to the large difference in the number of people included in the three areas, there is anobvious bias among them in the basic characteristics, such as age, education level, etc. Can the results of this manuscript really reflect the characteristics of the population?
  4. In conclusion, the author should give corresponding countermeasures and prospects.

8.The data of the percentage of the mother’s education level in the conclusion does not match the data provided in Table 1.

  1. A web-based questionnaire was used to collect detailed information. According to the research results of thismanuscript, the educational levels of the three areasare different, and the respondents' familiarity with computers may lead to the deviation of the information collected.

Author Response

Reviewer #2

Comments and suggestions for authors:

The authors created the Neonatal Environment and Health Outcomes (NEHO) longitudinal birth cohort in three ICSs in the Mediterranean area of southern Italy, collecting comprehensive information on personal data and lifestyles by questionnaire, to explore possible risk factors for negative pregnancy outcomes. The results suggested that SES influences both lifestyles known to produce negative pregnancy outcomes and a higher survey attrition rate.

R: We thank the Reviewer for her/his comments that helped us to improve the quality of our manuscript

Our comments on this article are as follows:

  1. The title of this article is not clear and specific, and readers don't know what the authors want to say? Abbreviations should not be used in the title, for even if an abbreviation is well known in one specialty, it could be confusing to readers from other specialties.

R: Also following the request from Reviewer #3, we changed the title of the manuscript into “The Neonatal Environment and Health Outcomes (NEHO) birth cohort study: behavioral and socioeconomic characteristics and drop-out rate from a longitudinal birth cohort in three industrially contaminated sites in southern Italy”

  1. The research goal of this paper is not clear.

R: The Reviewer is right. The aim of the work was not clearly presented to the reader. We changed the title and the last paragraph of Introduction section was entirely rewritten

  1. If the author wants to study the impact of environmental pollution on adverse pregnancy outcomes, there are serious defects in the design of this article. For example, since the study population comes from three high pollution areas, the environmental pollutants in the air, water, soil, and food should be the main risk factors for adverse pregnancy outcomes. This study did not involve the impact of environmental pollutants on adverse pregnancy outcomes. Obviously, the conclusion is not reliable.

R: We perfectly understand the Reviewer’s observation. Nevertheless, the aim of the present work was not to present results on the impact of environmental pollution on adverse pregnancy outcomes (e.g., malformations). As suggested by the Reviewer, at the end of Introduction section we now clearly state that the aim was to describe the behavioral, socioeconomic and gestational characteristics of the sample and their effect on drop-out rate. In a previous work (Ruggieri et al, 2019) we described the information needed for characterizing the sample: comprehensive information on pre-pregnancy health status (physical activity, lifestyle, stress factors, socio-demographic characteristics, use of medication, and information about previous births, including stillbirths). The same data are also collected for the gestational period, along with smoking habits, possible chemical exposures, and maternal health characteristics. After delivery, information is collected during children’s follow-up regarding use of medicine, nutritional outcomes (including growth and breastfeeding), neurocognitive development, infections and injuries, hospitalizations, and characteristics of home environments. In addition to the above-listed variables, the NEHO cohort collects detailed information about diet: the origin of the food (whether local or unknown) and place of purchase (local markets or large supermarket chains), as well as information about drinking water. Information on domestic and/or working environment exposures is also collected. Thus, we believe that – when information on exposure biomarkers will be available – we were able to define the impact of environmental pollution on children development in these areas.

The first paragraph of Conclusions section was accordingly modified.

The second paragraph does not refer to our study, but – in general – to the cohort studies. It was shortened as per following request (point 4) of the Reviewer

  1. Even if we don't talk about the impact of environmental pollutants on adverse pregnancy outcomes, the conclusions of this study are not innovative and belong to popular science knowledge.

R: The Conclusions section was modified and shortened, making it more relevant to the results of our manuscript, i.e., the effect of behavioral and socioeconomic factors on attrition rate.

  1. In the result section, each independent content should have a subtitle.

R: As requested, Results section was splitted into three subsections

  1. The NEHO cohort included 845 women, 561 (66.4%) from the Augusta-Priolo site, 167 (19.8%) from Crotone, and 117 (13.8%) from Milazzo. Due to the large difference in the number of people included in the three areas, there is an obvious bias among them in the basic characteristics, such as age, education level, etc. Can the results of this manuscript really reflect the characteristics of the population?

R: As concerns the education level we found that the relationship among the three evaluated sites were maintained. The site with the highest percentage of women with

  1. In conclusion, the author should give corresponding countermeasures and prospects.

R: The Conclusions section has been completely rewritten

  1. The data of the percentage of the mother’s education level in the conclusion does not match the data provided in Table 1.

R: We thank the Reviewer: an inconsistency was actually present between Table 1 and the text, due to rounding some values. This was now corrected in the Table

  1. A web-based questionnaire was used to collect detailed information. According to the research results of this manuscript, the educational levels of the three areas are different, and the respondents' familiarity with computers may lead to the deviation of the information collected.

R: The Reviewer is right: the educational level could have influenced the participation in the study of the recruited mothers. However, all the information on which the present work is based comes from the baseline questionnaire, collected through an interview carried out by an operator. Therefore, the participant’s familiarity with computers did not influence the response. This is now specified in 2.1. Study Population subsection.

Reviewer 3 Report

Summary

The authors present preliminary results on an interesting birth cohort from three industrially contaminated sites in southern Italy. Available covariates, inclusion criteria and biological samples are clearly outlined/stated to give reviewers an idea of where future data analysis will lead.

Major comments

From the title, it is unclear what first results are being presented. Please indicate that this is the methods paper or preliminary results on XYZ eg smoking, ses, etc. After reading through the paper, I can suggest that authors consider a title which contains a phrase like ‘characteristics of study population’ (as indicated in line 394).

As an example, please see PMID: 29547889, title and explanation for the manuscript is very clear: Cohort Profile: The Isle Of Wight Whole Population Birth Cohort (IOWBC)

Minor comments

  1. Introduction was longwinded. Lines 73 was pretty informative and helped reviewer see the gap in knowledge. Please consider summarizing the gaps in knowledge and going straight to the point of the need for the study. While truncating introduction, please reconsider the use of acronyms; there were quite a few that were confusing.
  2. What was the overall hypothesis for the study/manuscript presented? Or what specific questions were the authors hoping to answer? (I see a brief snippet of this answer on line 136, but this should have come earlier)
  3. Line 211: “relationship between “active smoking” during pregnancy and educational level was significant” not clear to me. Was this a stratification analysis?
  4. For table 1, although ns are presented in text on line 192, please present ns by site to make the table stand alone. I kept referring back to the text to see ns and other factors written in results (just ns by site at the very top will do). The same goes for any table where results are presented by site.
  5. Fig 2: What was the criteria for choosing the 6 variables for evaluation?

“Six variables relevant to gestational period and birth outcomes are evaluated”

  1. Line 118: why were pregnant women recruited during the 3rd trimester, as opposed to 1st or 2nd trimester?

  1. I did not see much on socio-economic factors – what factors exactly were included? Was this associated with the educational level? For instance, the authors did a great job analyzing smoking and education level, I am not clear on the socio-economic factors alluded to in the introduction.

Author Response

Reviewer #3

Summary

The authors present preliminary results on an interesting birth cohort from three industrially contaminated sites in southern Italy. Available covariates, inclusion criteria and biological samples are clearly outlined/stated to give reviewers an idea of where future data analysis will lead.

We thank the Reviewer for her/his comments which helped us improve the quality of the manuscript

Major comments

From the title, it is unclear what first results are being presented. Please indicate that this is the methods paper or preliminary results on XYZ eg smoking, ses, etc. After reading through the paper, I can suggest that authors consider a title which contains a phrase like ‘characteristics of study population’ (as indicated in line 394).

As an example, please see PMID: 29547889, title and explanation for the manuscript is very clear: Cohort Profile: The Isle Of Wight Whole Population Birth Cohort (IOWBC)

R: Also following the suggestion from Reviewer #2 the title was changed into “The Neonatal Environment and Health Outcomes (NEHO) birth cohort study: behavioral and socioeconomic characteristics and drop-out rate from a longitudinal birth cohort in three industrially contaminated sites in southern Italy”

Minor comments

  1. Introduction was longwinded. Lines 73 was pretty informative and helped reviewer see the gap in knowledge. Please consider summarizing the gaps in knowledge and going straight to the point of the need for the study. While truncating introduction, please reconsider the use of acronyms; there were quite a few that were confusing.

R: Following Reviewer’s suggestion, the Introduction section was shortened and some references deleted

  1. What was the overall hypothesis for the study/manuscript presented? Or what specific questions were the authors hoping to answer? (I see a brief snippet of this answer on line 136, but this should have come earlier)

R: The last paragraph of the Introduction section was modified accordingly

  1. Line 211: “relationship between “active smoking” during pregnancy and educational level was significant” not clear to me. Was this a stratification analysis?

R: It is not a stratification analysis. A logistic regression model was constructed in which the response variable is active smoking during pregnancy and the explanatory variable is the mother's educational level. With the aim of evaluating whether the mother's study title influences in some way the exposure to active smoking and therefore whether the risk of smoking is different for different levels of education. We thank the Reviewer for the observation: the information was unclear. Now in Methods section (first paragraph of subsection 2.3. Statistics) the adopted regression methods are indicated and, throughout the Results section, odds ratios are clearly referred to the relevant models.

  1. For table 1, although ns are presented in text on line 192, please present ns by site to make the table stand alone. I kept referring back to the text to see ns and other factors written in results (just ns by site at the very top will do). The same goes for any table where results are presented by site.

R: As requested, in all the Tables, for each site, the sample size relevant to each categorical variable is indicated

  1. Fig 2: What was the criteria for choosing the 6 variables for evaluation? “Six variables relevant to gestational period and birth outcomes are evaluated”

R: We selected these variables (weight gain, gestational age at first ultrasound scan, gestational age at birth, father’s age, newborn’s length, newborn’s head circumference) because, in our opinion, they were suitable for describing both the gestational period and the birth outcome

  1. Line 118: why were pregnant women recruited during the 3rd trimester, as opposed to 1stor 2nd trimester?

R: Two reasons.

The enrollment was performed in public hospitals, that are contacted by pregnant women during the last weeks before the expected date for delivery.

Moreover, an inclusion criterion was the absence of any evident complications during pregnancy diagnosed previous to signing informed consent. Consequently, we preferred to enroll women at the end of pregnancy.

  1. I did not see much on socio-economic factors – what factors exactly were included? Was this associated with the educational level? For instance, the authors did a great job analyzing smoking and education level, I am not clear on the socio-economic factors alluded to in the introduction.

R: The available variables relevant to SES, present in “Baseline First Part” questionnaire were marital status, educational level and working condition. These variable were able to provide significant results in cluster and MCA analysis.

Round 2

Reviewer 2 Report

The authors have revised the manuscript according to the suggestion and there is no other concerns regarding the manuscript.